# Foxc1 establishes enhancer accessibility for craniofacial cartilage differentiation

**Pengfei Xu[†], Haoze V Yu[†], Kuo-Chang Tseng, Mackenzie Flath, Peter Fabian, Neil Segil, J Gage Crump***

Eli and Edythe Broad Center for Regenerative Medicine, Department of Stem Cell Biology and Regenerative Medicine, Keck School of Medicine, University of Southern California, Los Angeles, United States

**Abstract** The specification of cartilage requires Sox9, a transcription factor with broad roles for organogenesis outside the skeletal system. How Sox9 and other factors gain access to cartilage-specific cis-regulatory regions during skeletal development was unknown. By analyzing chromatin accessibility during the differentiation of neural crest cells into chondrocytes of the zebrafish head, we find that cartilage-associated chromatin accessibility is dynamically established. Cartilage-associated regions that become accessible after neural crest migration are co-enriched for Sox9 and Fox transcription factor binding motifs. In zebrafish lacking Foxc1 paralogs, we find a global decrease in chromatin accessibility in chondrocytes, consistent with a later loss of dorsal facial cartilages. Zebrafish transgenesis assays confirm that many of these Foxc1-dependent elements function as enhancers with region- and stage-specific activity in facial cartilages. These results show that Foxc1 promotes chondrogenesis in the face by establishing chromatin accessibility at a number of cartilage-associated gene enhancers.

**\*For correspondence:**
gcrump@usc.edu

[†]These authors contributed equally to this work

**Competing interests:** The authors declare that no competing interests exist.

## Introduction

Cartilage is the first skeletal type to be specified in the vertebrate body, providing important templates for later bone development and providing flexibility at joint surfaces and within the nose, ear, ribs, and larynx. The transcription factor Sox9 is essential for chondrogenic differentiation in all vertebrates examined (*Bi et al., 1999*; *Lefebvre et al., 1997*; *Mori-Akiyama et al., 2003*; *Yan et al., 2005*), yet it also has widespread roles outside the skeletal system, including the reproductive system, kidney, liver, and skin (*Jo et al., 2014*). How Sox9 is directed to a chondrogenic program in trunk mesoderm and cranial neural crest-derived cells (CNCCs) has remained unclear. Sox9 is known to directly bind to a number of cis-regulatory elements adjacent to chondrogenic genes, including *Col2a1*, *Col10a1*, and *Acan* (*Askary et al., 2015*; *Dy et al., 2012*; *Lefebvre et al., 1997*; *Ohba et al., 2015*). It is, however, dispensable for chromatin accessibility at these same elements (*Liu et al., 2018*), suggesting that other unknown factors may first open chromatin at chondrogenic enhancers for later activation by Sox9.

Forkhead-domain (Fox) family transcription factors are excellent candidates for establishing chromatin accessibility at chondrogenic enhancers. In the endoderm lineage, HNF3/FoxA binds closed chromatin at enhancers and makes these more accessible (*Cirillo et al., 2002*). Foxd3 has been similarly proposed to establish chromatin accessibility in the early neural crest lineage (*Lukoseviciute et al., 2018*). In mouse, loss of *Foxc1* results in widespread cartilage and bone defects (*Kume et al., 1998*), including impaired tracheal and rib cartilages (*Hong et al., 1999*), loss of calvarial bone due to premature ossification (*Rice et al., 2003*; *Sun et al., 2013*; *Vivatbutsiri et al., 2008*), syngnathia (*Inman et al., 2013*), and disruption of endochondral bone maturation (*Yoshida et al., 2015*). In zebrafish, loss of both Foxc1 paralogs (*foxc1a* and *foxc1b*) results in severe reductions of dorsal cartilages of the upper face, which are preceded by reduced

**eLife digest** Animals with backbones (or vertebrates) have body shape determined, in part, by their skeletons. These emerge in the embryo in the form of cartilage structures that will then get replaced by bone during development. The neural crest is a group of embryonic cells that can become different tissues. In the head, it forms the cartilage scaffold for some of the facial bones and the base of the skull.

During this process, a protein called Sox9 is required for neural crest cells to morph into cartilage. This transcription factor binds to regulatory sequences in the genome to turn cartilage genes on. But Sox9 is also required to form non-cartilage tissues in organs such as the liver, lung, and kidneys. How, then, does Sox9 only turn on the genes required for cartilage formation in the embryonic face? This specificity can be controlled by which regulatory sequences Sox9 can physically access in a cell: controlling which regulatory sequences Sox9 can access determines which genes it can activate, and which type of tissue a cell will become.

Xu, Yu et al. wanted to understand exactly how Sox9 switches on the genes that turn neural crest cells into facial cartilage. They studied the genomes of zebrafish embryos, which have a cartilaginous skeleton similar to other vertebrates, and found out which areas were accessible to transcription factors in the neural crest cells that became facial cartilage. Analyzing these regions suggested that sites where Sox9 could bind were often close to binding sites for another protein, called Foxc1. When zebrafish embryos were genetically modified to inactivate Foxc1 proteins, many of the regulatory sequences in cartilage failed to become accessible, and the cartilaginous skeleton did not form properly. These results support a model in which Foxc1 opens up the genomic regions that Sox9 needs to bind for cartilage to form, as opposed to the regions that Sox9 would bind to make different organ cell types.

The findings of Xu, Yu et al. uncover the stepwise process by which cartilage cells are made during development. Further research based on these results could allow scientists to develop new ways of replacing cartilage in degenerative conditions such as arthritis.

expression of several Sox9 targets, including *col2a1a*, *acana*, *matn1*, and *matn4* (*Xu et al., 2018*). Chromatin immunoprecipitation followed by deep sequencing (ChIP-Seq) using a Sox9 antibody in dissected mouse rib and nose cartilage revealed enrichment of Fox binding motifs within Sox9-bound cis-regulatory sequences near chondrogenic genes (*Ohba et al., 2015*). Here, we use profiling of chromatin accessibility in wild-type and mutant zebrafish facial cartilages and find that Foxc1 paralogs are required for accessibility and activity of a number of cartilage enhancers. These findings support a model in which Foxc1 promotes chondrogenesis by establishing selective accessibility of cartilage enhancers in CNCC-derived mesenchyme.

## Results and discussion

### Chromatin accessibility landscape in chondrocytes of the zebrafish face

In order to identify potential cis-regulatory elements important for facial cartilage development, we performed a genome-wide analysis of chromatin accessibility in chondrocytes from 72 hr post-fertilization (hpf) zebrafish. We labeled chondrocytes by co-expression of *sox10:Dsred* and *col2a1a:GFP* transgenes (*Figure 1A*) and isolated double-positive and control double-negative cells by fluorescence-activated cell sorting (FACS). We then subjected these cells to a modified 'micro' version of the assay for transposase-accessible chromatin followed by next-generation sequencing (μATACseq). In order to focus on potential distal cis-regulatory elements, we excluded accessible regions within 1 kb upstream or 0.5 kb downstream of transcription start sites. This analysis yielded 33,679 distal accessible elements, with 5736 elements over-enriched in chondrocytes and 8955 elements under-enriched in chondrocytes (*Figure 1B,C*). As confirmation of sequence quality, the cartilage-specific R2 enhancer of the Collagen Type II alpha 1 a gene (*col2a1a*) is enriched in chondrocyte-specific elements (*Figure 1—figure supplement 1A*). De novo motif analysis of the top 2000 chondrocyte-enriched regions using HOMER recovered motifs for Sox, Fox, Nfat, Zfx, and Nkx transcription factor families (*Figure 1D*; *Supplementary file 1A*). Sox9 ChIP-Seq of mouse chondrocytes had previously

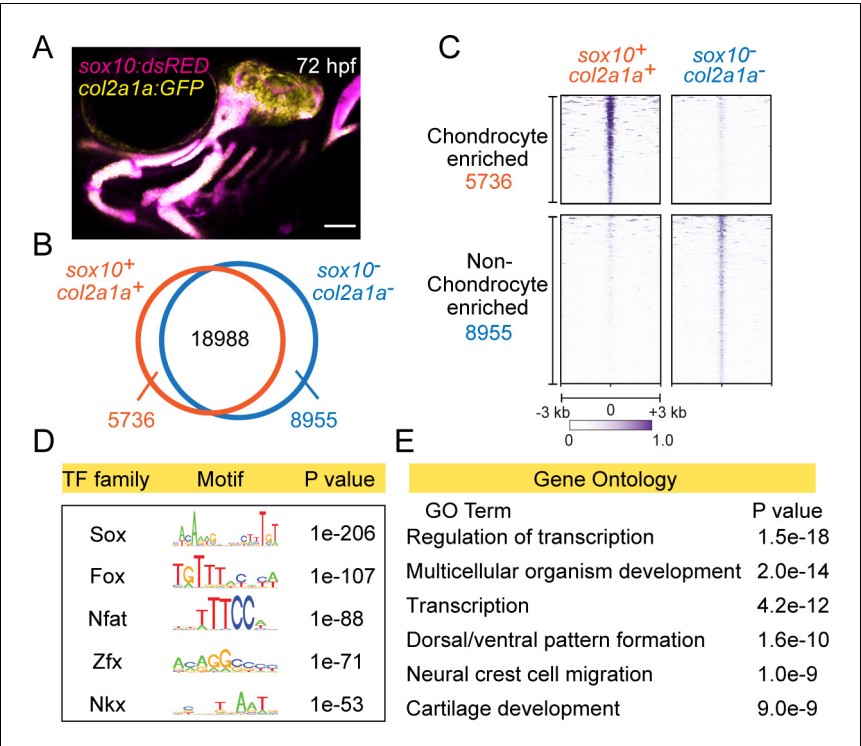

**Figure 1.** Chromatin accessibility landscape of facial chondrocytes. (**A**) Confocal image of facial cartilages expressing *col2a1a:GFP* and *sox10:Dsred* at 72 hpf. Lateral view with anterior to left. Scale bar = 100 μm. (**B**) Venn diagram indicating distal elements with open chromatin accessibility in *col2a1a:GFP+; sox10:Dsred+* versus *col2a1a:GFP−; sox10:Dsred−* cells. (**C**) Peak intensity plots (heatmap) of μATACseq show differentially enriched open chromatin regions in double-positive versus double-negative cells. (**D**) The top five transcription factor (TF) motifs recovered from the top 2000 μATACseq peaks enriched in chondrocytes (after removing redundant motifs). (**E**) GO analysis of nearest neighbor genes of μATACseq peaks enriched in chondrocytes.
The online version of this article includes the following figure supplement(s) for figure 1:

**Figure supplement 1.** *col2a1a* enhancer and cartilage motif comparison between zebrafish and mouse.

revealed Nfat and Fox motifs as the second and third most co-enriched with Sox motifs (*Ohba et al., 2015*), and the Nkx motif might reflect the role of Nkx3.2 in cartilage differentiation (*Provot et al., 2006*). Consensus sequences for Sox, Fox, and Nfat motifs were highly similar between zebrafish and mouse (*Figure 1—figure supplement 1B*), despite our zebrafish analysis focusing on all accessible regions in facial chondrocytes and mouse analysis focusing on only Sox9-bound regions in rib chondrocytes. These striking motif similarities indicate strong conservation of the cartilage gene regulatory network between fish and mammals, and between the face and rib, and strongly suggest that many of the identified chondrocyte-specific accessible regions in zebrafish likely function as chondrocyte enhancers. In addition, Gene Ontology (GO) analysis of the nearest genes to the chondrocyte-enriched elements revealed cartilage development as one of the top six associated terms (*Figure 1E*). We also recovered terms for neural crest cell migration and dorsal/ventral pattern formation, likely reflecting retention of enhancer accessibility linked to the neural crest origins and later dorsoventral arch patterning of the precursors of facial cartilage.

## Progressive establishment of cartilage-associated chromatin accessibility

We next investigated when cartilage-associated chromatin accessibility is established in relation to CNCC development. CNCCs are first specified at ~10.5 hpf at the border of the neural keel, finish their migration into the pharyngeal arches by ~20 hpf, and then show the first histological signs of cartilage development in the jaw at ~52 hpf (*Schilling and Kimmel, 1997*). CNCC-derived arch

ectomesenchyme cells can be uniquely identified by co-expression of *fli1a:GFP* and *sox10:Dsred* transgenes at 36 and 48 hpf (*Askary et al., 2017*), stages just prior to cartilage differentiation (*Figure 2A*). We therefore performed μATACseq on FACS-purified *fli1a:GFP+; sox10:Dsred+* cells at these stages. 4915 of 14,623 elements gaining accessibility from 36 to 48 hpf were linked to GO terms including skeletal system development and cartilage development, and were enriched for Sox and Fox transcription factor binding motifs. 4323 elements with decreasing accessibility were linked to several GO terms related to cellular migration and enriched for motifs of neural crest-associated transcription factors (Nr2f, Lhx, Olig2, Hox, Ets), consistent with decommissioning of enhancers involved in earlier neural crest migration (*Figure 2—figure supplement 1*).

To more specifically understand the timing of cartilage-associated chromatin accessibility, we next analyzed chondrocyte-specific accessible elements from our 72 hpf dataset for their accessibilities at 36 and 48 hpf (*Figure 2B*). Of the 5,736 elements enriched in chondrocytes at 72 hpf, only 6% (356) had peak accessibility at 36 hpf with no further increase in accessibility by 48 hpf ('Group I'). In contrast, 48% (2741) displayed increased accessibility between 36 and 48 hpf ('Group II'), and 46% (2639) between 48 and 72 hpf ('Group III'). For Group I, de novo motif enrichment revealed predicted binding sites for members of the Nfat, Fox, Lhx, Nr2f, Meis, and Pax families, and significant GO terms included neural crest migration, cell migration in general, and dorsal/ventral pattern formation (*Figure 2C,D*). Combined with the known involvement of Foxd3 (*Montero-Balaguer et al., 2006*; *Stewart et al., 2006*), Lhx6/8 (*Denaxa et al., 2009*), Nr2f1/2/5 (*Barske et al., 2018*), Meis2 (*Machon et al., 2015*), and Pax9 (*Nakatomi et al., 2010*) in CNCC specification, migration, and dorsal–ventral arch patterning, many Group I elements likely represent retention of cis-regulatory elements involved in the earlier specification, migration, and regional patterning of CNCCs. Group II and Group III elements share many common predicted transcription factor binding motifs, including Sox dimer, Fox, Nfat, and Ap1 motifs previously described for mouse cartilage (*Figure 2E,G*; *Supplementary file 1B*; *Ohba et al., 2015*). An Nkx motif was recovered only for Group II (p=$10^{-58}$, 36% of targets), an Egr motif was enriched for Group II (p=$10^{-65}$, 55% of targets) versus Group III (p=$10^{-20}$, 16% of targets), and a Tead motif was enriched for Group III (p=$10^{-63}$, 36% of targets) versus Group II (p=$10^{-20}$, 3% of targets). GO analysis for linked genes also revealed terms related to skeletal system development (Group II) and cartilage development (Group III), as well as more general terms such as transcription and cell differentiation (*Figure 2F,H*). We therefore conclude that the majority of chondrocyte-specific elements gain accessibility after pharyngeal arch formation and that transcription factor binding motifs change during cartilage differentiation. For example, enrichment of Nkx motifs in Group II elements might reflect the role of Nkx3.2 in limiting chondrocyte maturation (*Provot et al., 2006*) and promoting joint formation (*Miller et al., 2003*), while the preferential enrichment of the Tead motif, which is linked to growth-associated Hippo signaling (*Ota and Sasaki, 2008*), in Group III elements might reflect the later proliferative expansion of chondrocytes.

## Requirement of Foxc1 for chromatin accessibility at a subset of cartilage elements

We had previously found that Foxc1 genes are essential for cartilage development in the upper face (*Xu et al., 2018*), and both our μATACseq analysis of zebrafish chondrocytes and published Sox9 ChIP-seq analysis in mouse (*Ohba et al., 2015*) reveals co-enrichment of Sox and Fox motifs in accessible regions near known cartilage genes. In zebrafish Foxc1 (*foxc1a−/−; foxc1b−/−*) mutants, cartilages of the upper/dorsal face fail to develop (*Xu et al., 2018*). In order to isolate the dorsal arch CNCC precursors affected in Foxc1 mutants, we used a *pou3f3b:Gal4; UAS:nlsGFP* (*pou3f3b>GFP*) dorsal CNCC transgenic line (*Barske et al., 2020*) along with the pan-CNCC *sox10:Dsred* transgenic line. In situ hybridization for *foxc1a* and *foxc1b* at 36 hpf showed partial overlap with the *pou3f3b>GFP* line in the dorsal-intermediate regions of the first and second arches (*Figure 3—figure supplement 1A*). Consistently, we observed reductions of cartilage in the *pou3f3b>GFP* domain of mutants at 6 days post-fertilization (dpf), except in the dorsal-most regions (close to the ear) that are *pou3f3b>GFP*-positive yet *foxc1a/foxc1b*-negative (*Figure 3C,D*; *Figure 3—figure supplement 1B*). As *pou3f3b>GFP+; sox10:Dsred+* CNCCs were present at 48 hpf (*Figure 3A,B*), we performed μATACseq on these cells after FACS from Foxc1 mutants and controls at 36 and 48 hpf. A comparison of the 15,781 accessible regions in dorsal CNCCs (*pou3f3b>GFP+; sox10:Dsred+*) and pan-CNCCs (*fli1a:GFP+; sox10:Dsred+*) at 36 hpf revealed 79% with similar

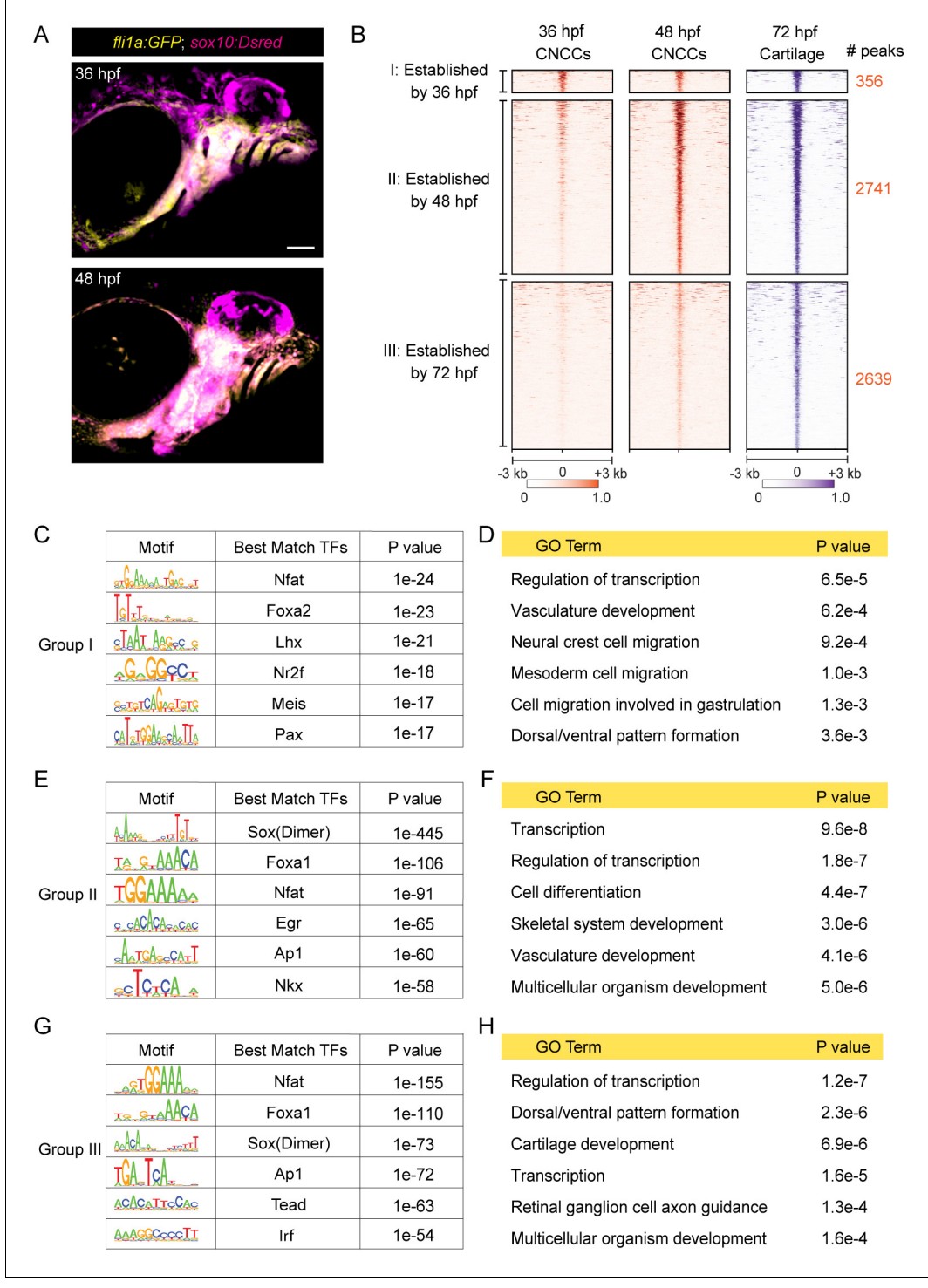

**Figure 2.** Dynamics of chromatin accessibility across facial chondrogenesis. (**A**) Confocal images of CNCCs expressing *fli1a:GFP* and *sox10:Dsred* at 36 and 48 hpf. Lateral view with anterior to left. Scale bar = 100 μm. (**B**) Peak intensity plots of cartilage-accessible distal elements shown for chondrocytes at 72 hpf and CNCCs at 36 and 48 hpf. Chondrocyte accessible elements are pooled into three categories based on dynamics of chromatin accessibility across stages. (**C, E, G**) De novo motif enrichment recovered by Homer analysis among the three categories. Top six motifs are shown with associated p values after removing redundant motifs. (**D, F, H**) GO term analysis among the three categories.

The online version of this article includes the following figure supplement(s) for figure 2:

*Figure 2 continued on next page*

*Figure 2 continued*

**Figure supplement 1.** Comparison of accessible regions between 36 and 48 hpf CNCCs.

accessibility, 11% with greater accessibility in dorsal CNCCs, and 10% with greater accessibility in pan-CNCCs (*Figure 3—figure supplement 2A*). Of 22,323 elements at 48 hpf, 72% were similarly accessible, 10% more accessible in dorsal CNCCs, and 18% more accessible in pan-CNCCs (*Figure 3—figure supplement 2B*). A comparison of the 5,736 regions with specific accessibility in cartilage at 72 hpf revealed high correlation between dorsal- and pan-CNCCs at 36 hpf, with 96% displaying similar accessibility (r = 0.92, *Figure 3—figure supplement 3A,C*). By 48 hpf, however, we observed notable differences between accessibility of cartilage-specific elements, with 43% displaying greater accessibility in pan-CNCCs and only 0.2% displaying greater accessibility in dorsal CNCCs (r = 0.71, *Figure 3—figure supplement 3B,D*). The decreased accessibility of cartilage-specific elements in dorsal CNCCs at 48 hpf supports previous studies that dorsal chondrocytes develop later than other chondrocytes in the zebrafish face (*Barske et al., 2016*; *Schilling and Kimmel, 1997*).

Analysis of cartilage-associated elements in Foxc1 mutants revealed that 10% (120/1221) of elements that have established peak accessibility by 36 hpf (i.e. Group I) and 41% (636/1556) of elements that increase accessibility between 36 and 48 hpf (i.e. Group II) had reduced accessibility in Foxc1 mutants (*Figure 3E*, *Figure 3—figure supplement 4*). De novo motif analysis of Foxc1-dependent and Foxc1-independent elements in Group I showed enrichment of Sox ($p=10^{-12}$, $p=10^{-76}$), Tead ($p=10^{-14}$, $p=10^{-20}$), Mtf ($p=10^{-12}$, $p=10^{-22}$), and Ets ($p=10^{-12}$, $p=10^{-61}$) motifs in both (*Figure 3F*, *Supplementary file 1C*). Whereas several types of Fox motifs were uncovered in both, a closer analysis revealed enrichment of Foxa2 and Foxd3 motifs only in Foxc1-dependent elements ($p=10^{-45}$, 38% of targets; $p=10^{-16}$, 18% of targets), and a Foxo1 motif only in Foxc1-independent elements ($p=10^{-42}$, 49% of targets). Similarly in Group II elements, a Foxa1 motif was enriched only in Foxc1-dependent elements ($p=10^{-59}$, 38% of targets), and Foxh1 and Foxp1 motifs only in Foxc1-independent elements ($p=10^{-22}$, 22% of targets; $p=10^{-19}$, 7% of targets) (*Figure 3G*, *Supplementary file 1C*). Although Foxc1 motifs were not in the database used for motif predictions, the selective presence of Foxa1/2 motifs in Foxc1-dependent elements is consistent with previous reports that the consensus sequence for Foxc1-bound peaks is nearly identical to Foxa1/2 motifs (*Wang et al., 2016*). Sox motifs were similarly enriched in both Group I and Group II Foxc1-dependent and -independent elements, and Nfat and Zfx motifs in Group II Foxc1-dependent and -independent elements. An Insm ($p=10^{-39}$, 30% of targets) motif was uncovered only in Foxc1-independent Group I elements and an Nkx motif ($p=10^{-35}$, 52% of targets) only in Foxc1-independent Group II elements. Whereas 33% (207/636) of Foxc1-dependent Group II elements had both Sox and Fox predicted motifs, only 16% (151/920) of Foxc1-independent Group II elements had both. These findings suggest that Foxc1-dependent and -independent cartilage elements may be commonly bound by Sox9 but likely differ in co-binding by Foxc1 and additional co-factors. For example, the presence of the Nkx motif only in Foxc1-independent elements suggests that it could be an alternative co-factor for Sox9, in line with the known roles for Nkx3.2 in chondrocyte biology (*Provot et al., 2006*).

## Validation of Foxc1-dependent cartilage enhancers in zebrafish transgenesis assays

To verify whether Foxc1-dependent cartilage elements identified by µATACseq are chondrogenic enhancers, we tested the ability of individual elements in combination with an E1b minimal promoter to drive cartilage expression of green fluorescent protein (GFP) in zebrafish transgenic assays (*Figure 4—figure supplement 1A*). We tested 22 Foxc1-dependent Group II elements near 15 different genes, which included elements linked to genes with known cartilage function (*ucmab*, *matn4*, *matn1*, *lect1*, *epyc*, *col9a1a*, *col9a3*, *sox10*, *acana*, *foxa3*, *mia*) and others with unknown cartilage function (*si:dkey33i1l.4*, *gas1b*, *lefty2*, *slc35d1a*) (*Supplementary file 2*). We observed that 59% (13/22) of elements drove GFP expression in facial chondrocytes at 6 dpf. These included intronic elements within *sox10*, *lect1*, *col9a3*, *col9a1a*, and *slc35d1a*; distal 5′ elements near *ucmab*, *epyc*, *mia*, *acana*, and *matn4*; a distal 3′ element near *gas1b*; and a promoter-associated element for *lect1*. We

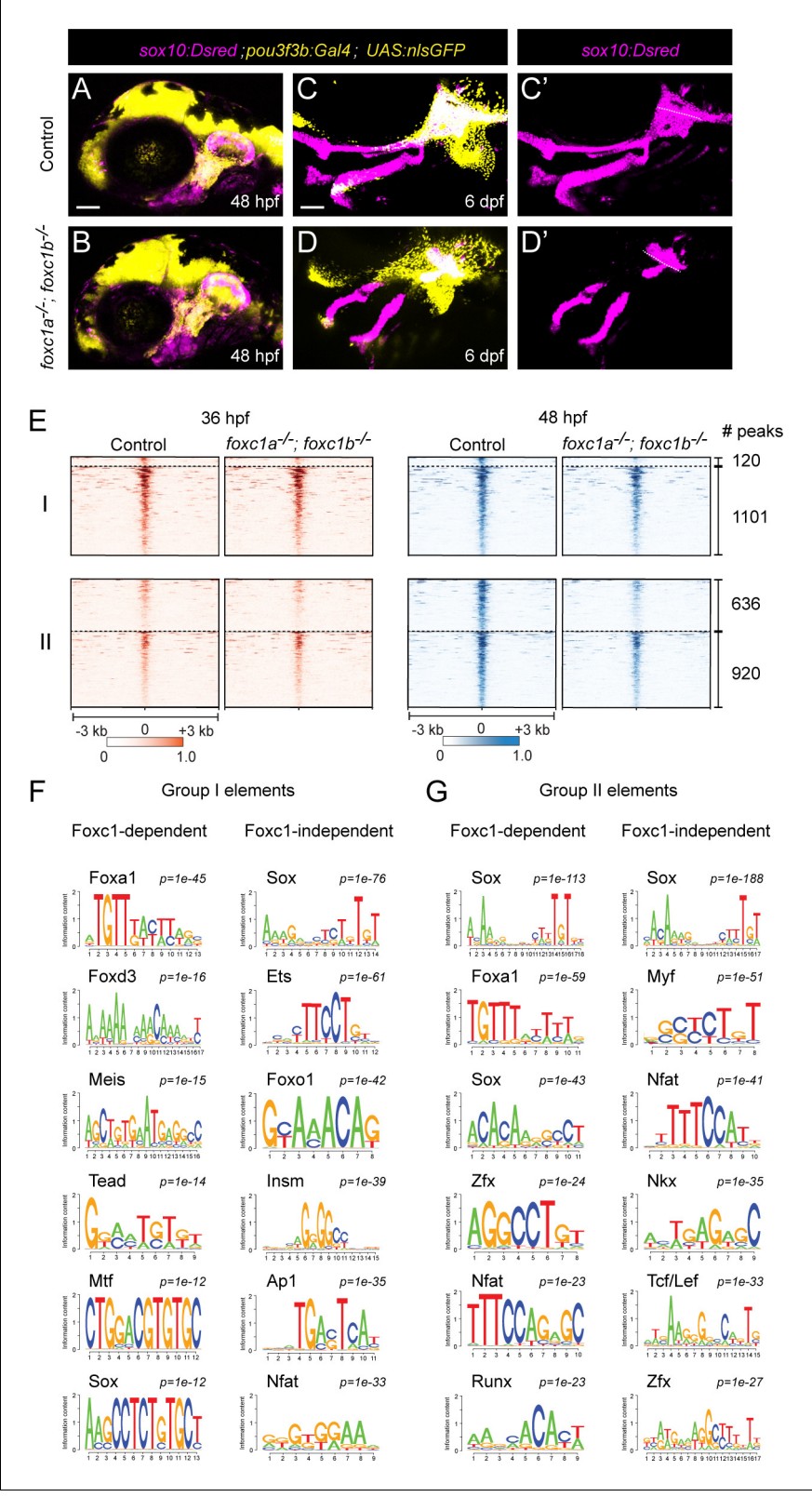

**Figure 3.** Foxc1 dependency of facial chondrocyte chromatin accessibility. (**A,B**) Confocal images show dorsal CNCCs of the first two arches labeled by *sox10:Dsred* and *pou3f3b:Gal4; UAS:nlsGFP* in control and *foxc1a*^−/−^; *foxc1b*^−/−^ mutant embryos at 48 hpf. Scale bar = 100 μm. (**C,D**) Confocal images show loss of dorsal cartilages in *foxc1a*^−/−^; *foxc1b*^−/−^ mutant embryos at 6 dpf. *sox10:Dsred+* cartilages are seen in single channels in **C'** and **D'**, *Figure 3 continued on next page*

*Figure 3 continued*

with dashed lines highlighting boundaries of dorsal arch and otic cartilage. (E) Peak intensity plots of Group I and Group II elements in control and *foxc1a*$^{-/-}$; *foxc1b*$^{-/-}$ mutant embryos. Peaks above the dashed lines are reduced in mutants. (F,G) De novo motif enrichment of Foxc1-dependent and Foxc1-independent Group I and Group II elements. Top six motifs are shown with associated p-values after removing redundant motifs.

The online version of this article includes the following figure supplement(s) for figure 3:

**Figure supplement 1.** Overlapping expression of *foxc1a* and *foxc1b* with *pou3f3b>GFP* in the dorsal-intermediate arches.

**Figure supplement 2.** Comparison of accessible regions between pan- and dorsal CNCCs.

**Figure supplement 3.** Accessibility of cartilage-enriched elements in earlier pan- and dorsal CNCCs.

**Figure supplement 4.** Volcano plots for differentially accessible peaks in Foxc1 mutants.

confirmed cartilage expression in independent stable transgenic lines for all 13 positive elements (*Figure 4A*; *Figure 4—figure supplement 1*; *Supplementary file 2*). Whereas an element in the first intron of *sox10* drove uniform cartilage-specific expression, most elements drove expression in specific sub-regions or particular differentiation stages of cartilage. A distal 5′ element of *ucmab* drove expression in chondrocytes of multiple joints in the zebrafish head, an intronic element of *lect1* drove chondrocyte expression only in the jaw joint and hyomandibular-otic connection, a promoter-associated element of *lect1* drove expression more strongly in the hyoid joint and hyomandibular-symplectic connection (though also more broadly in chondrocytes), a distal 5′ element of *acana* drove restricted expression at the hyoid joint, and a distal 3′ element of *gas1b* drove restricted expression at the Meckel's–Meckel's joint and connection between the hyomandibular cartilage and opercular bone. Reciprocally, elements associated with *col9a3*, *epyc*, *mia*, *matn4*, and *col9a1a* were expressed in chondrocytes but generally excluded from joint regions (particularly apparent at the hyoid joint and hyomandibular-symplectic connection). An intronic element for *slc35d1a* drove expression in both pharyngeal cartilage and muscle. Furthermore, we found that three enhancer transgenes with diverse expression patterns (broad *sox10*, joint-restricted *ucmab*, and joint-excluded *epyc* enhancers) all displayed reduced activity specifically in the dorsal cartilage regions affected in Foxc1 mutants (*Figure 4B*). We also confirmed by in situ hybridization that four genes linked to Foxc1-dependent enhancers (*sox10*, *lect1*, *col9a3*, *epyc*) showed reduced expression in cartilage-forming regions of the dorsal arches of Foxc1 mutants (*Figure 4C*), similar to our previous results for *col2a1a*, *matn1*, *matn4*, and *acana* (*Xu et al., 2018*).

Testing of four Foxc1-independent Group II elements revealed two that drove cartilage expression (distal 5′ elements of *gas1b* and *matn4*), one that drove cartilage, bone, and ligament expression (distal 5′ element of *sparc*), and one with no activity (promoter-associated element of *sox10*) (*Figure 4—figure supplement 1*). Thus, the majority of tested Foxc1-dependent and -independent Group II elements are equally capable of driving cartilage expression. We also tested five Foxc1-independent Group I elements and found that two showed arch CNCC expression at 36 hpf and minimal cartilage expression at 6 dpf (*prrx1a* and *emx3* elements), one no expression at 36 hpf and ligament expression at 6 dpf (*satb2*), and two no expression at either stage (*prrx1a*, *cd248a*) (*Figure 4—figure supplement 2*). In contrast, we did not observe 36 hpf arch CNCC expression in any of the 16 Group II elements that drove cartilage expression at 6 dpf. As most of the Group I elements tested had decreased accessibility from 36 to 48 hpf, it seems likely that these elements represent either neural crest elements in the process of decommissioning (e.g. *prrx1a* and *emx3*) or elements prefiguring non-cartilage fates (e.g. *satb2*). Thus, most cartilage enhancers appear to gain accessibility after 36 hpf, with activity of late-opening cartilage enhancers in diverse locations, indicating that global cartilage expression patterns are achieved in part through the summation of enhancers with more restricted activity.

## Conclusion

Our findings indicate that Foxc1 function is required for the accessibility of close to half of chondrocyte enhancers in the zebrafish face. Given expression and function of Foxc1 in diverse cartilages (e.g. limb, rib, tracheal) in mouse, it seems likely that Foxc1 has a similar function in chondrocyte enhancer accessibility throughout the body. The co-enrichment of predicted Sox and Fox binding sites in 33% of chondrocyte enhancers suggests a model in which Foxc1 promotes Sox9 binding to

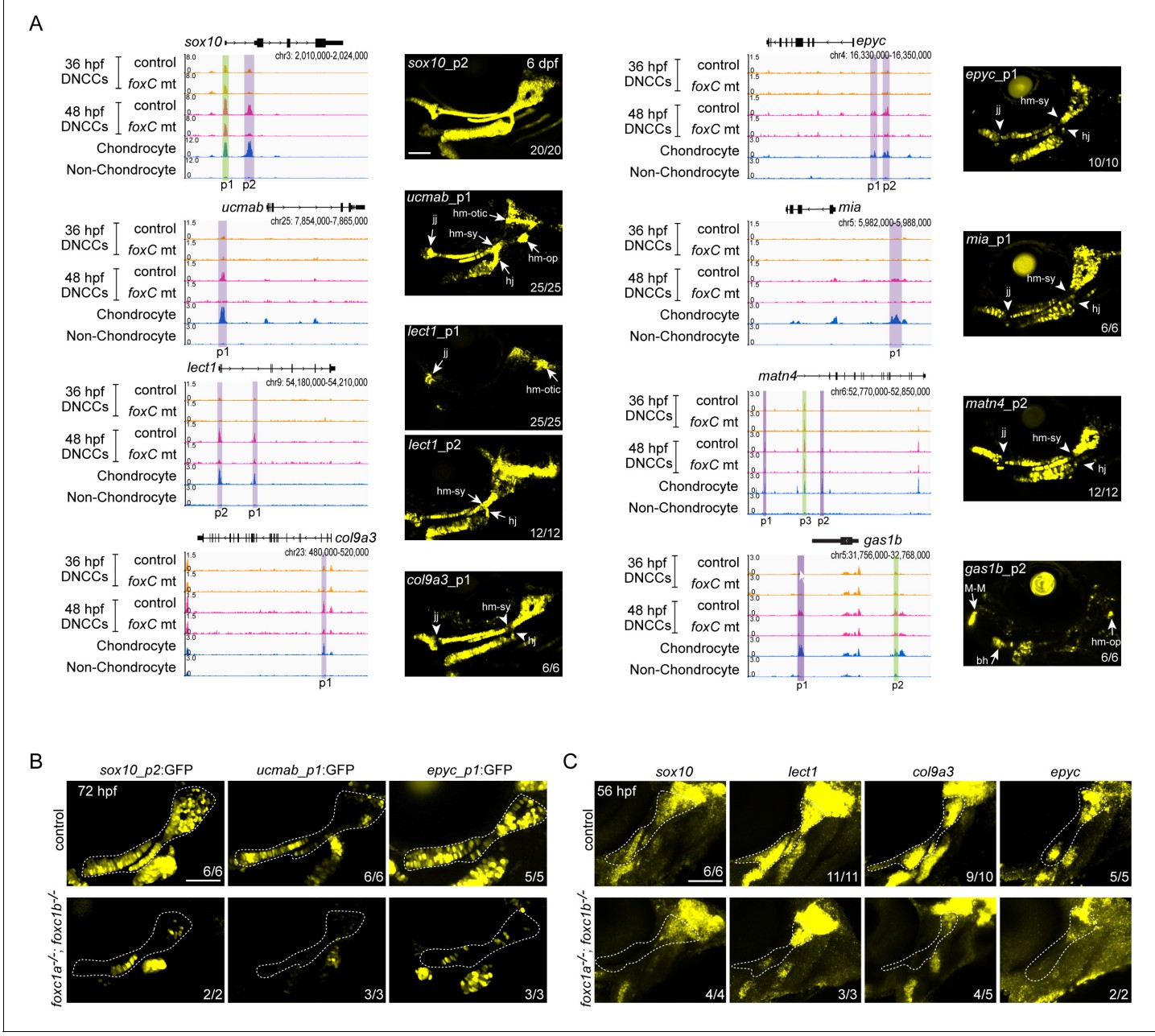

**Figure 4.** In vivo validation of Foxc1-dependent cartilage enhancers. (**A**) Genomic regions (gene loci and GRCz10 coordinates listed) for enhancer testing on the left and GFP expression driven by the indicated peaks in stable transgenic zebrafish at 6 dpf on the right. Peaks (p) tested are shown, with Foxc1-dependent regions in purple and Foxc1-independent elements in green. μATACseq reads are shown in each row, with chondrocyte and non-chondrocyte peaks from 72 hpf embryos. Confocal projections show cartilages of the first two arches in lateral view with anterior to the left. Arrows indicate enriched expression at joint regions, and arrowheads denote relative lack of expression. (**B**) Confocal projections show selective loss of *sox10_p2:EGFP*, *ucmab_p1:EGFP*, and *epyc_p1:EGFP* transgene expression in the dorsal cartilage domains (dashed outlines) of *foxc1a−/−; foxc1b−/−* mutants at 72 hpf. (**C**) Confocal projections of in situ hybridization show selective loss of *sox10*, *lect1*, *col9a3*, and *epyc* in the dorsal cartilage domain (dashed outline) of *foxc1a−/−; foxc1b−/−* mutants at 56 hpf. Numbers indicate proportion of embryos in which the displayed patterns were observed. bh, basihyal; DNCCs, dorsal CNCCs; jj, jaw joint; hj, hyoid joint; hm-op, hyomandibular-opercular joint; hm-otic, hyomandibular-otic junction; hm-sy, hyomandibular-symplectic junction; M-M, Meckel's–Meckel's joint. Scale bars = 100 μm.

The online version of this article includes the following figure supplement(s) for figure 4:

**Figure supplement 1.** Additional in vivo validation of cartilage-enriched accessible elements.

**Figure supplement 2.** In vivo validation of accessible elements from Group I.

these same enhancers by increasing chromatin accessibility, though this would require direct Sox9 binding studies to validate. However, not all Foxc1-dependent enhancers have predicted Fox binding motifs, which might suggest indirect regulation of chromatin accessibility in at least some cases. In addition, many enhancers do not appear to require Foxc1 activity. It is possible that other members of the Fox family compensate, such as Foxf1/2 in the facial midline (*Xu et al., 2018*) and Foxa2/3 during later hypertrophic maturation (*Ionescu et al., 2012*). Alternatively, there may be other co-factors that mediate chondrocyte enhancer accessibility and activation, such as Nkx3.2. We did not detect any obvious differences in the types of enhancer-proximal genes or the patterns and stages associated with Foxc1-dependent versus -independent enhancer activity. Further work will also be needed to understand the mechanism by which Foxc1 promotes chondrocyte enhancer accessibility, and the extent to which this reflects direct binding of Foxc1 to chondrogenic enhancers. Foxc1 lacks a chromatin modifying domain (*Yoshida et al., 2015*) and therefore would need to interact with a co-factor to directly open chromatin. Foxc1 could also act to maintain open chromatin, as shown for Foxa1 in the liver (*Reizel et al., 2020*). Given that both Foxc1 and Sox9 have expression in many tissues outside the skeletal system, it will also be important to determine whether additional factors help to further restrict their activity to chondrocyte enhancers within skeletogenic mesenchyme.

## Materials and methods

### Key resources table

| Reagent type (species) or resource | Designation | Source or reference | Identifiers | Additional information |
|---|---|---|---|---|
| Genetic reagent (*D. rerio*) | col2a1aBAC:GFP*el483* | PMID:26555055 | RRID:ZFINID:ZDB-ALT-160204-6 | Available at ZIRC |
| Genetic reagent (*D. rerio*) | fli1a:eGFP*y1* | PMID:12167406 | RRID:ZFINID:ZDB-ALT-011017-8 | Available at ZIRC |
| Genetic reagent (*D. rerio*) | sox10:Dsred*el10* | PMID:22589745 | RRID:ZFINID:ZDB-ALT-120523-6 | |
| Genetic reagent (*D. rerio*) | pou3f3b*Gal4ff-el79* | PMID:32958671 | | |
| Genetic reagent (*D. rerio*) | UAS:nlsGFP;α-crystallin:Cerulean*el609* | PMID:32958671 | | |
| Genetic reagent (*D. rerio*) | foxc1a*el543* | PMID:29777011 | RRID:ZFINID:ZDB-ALT-190103-11 | |
| Genetic reagent (*D. rerio*) | foxc1b*el620* | PMID:29777011 | RRID:ZFINID:ZDB-ALT-190104-4 | |
| Antibody | Rabbit polyclonal anti-GFP | Torrey Pines Biolabs | RRID:AB_10013661 | Used at 1:1000 |
| Antibody | Goat polyclonal anti-rabbit Alexa Fluor 568 | Thermo Fisher Scientific | RRID:AB_143157 | Used at 1:500 |

### Zebrafish lines

The Institutional Animal Care and Use Committee of the University of Southern California approved all experiments on zebrafish (*Danio rerio*) (Protocol #10885). Existing mutant and transgenic lines used in this study include *foxc1a*el542* and *foxc1b*el620* (*Xu et al., 2018*); *Tg(sox10:Dsred)*el110* and *Tg(fli1a:EGFP)*y1* (*Askary et al., 2017*); *Tg(col2a1aBAC:GFP)*el483* (*Paul et al., 2016*); and *Tg(UAS:nlsGFP;α-crystallin:Cerulean)*el609* and *pou3f3b*Gal4ff-el79* (*Barske et al., 2020*). For enhancer transgenic lines, we synthesized accessible elements with flanking attB4 and attB1 sequences using IDT gBlocks and cloned these into pDONR-P4-P1R using the Gateway Tol2kit (Invitrogen) to create p5E enhancer constructs (*Kwan et al., 2007*). We then combined p5E constructs with pME-E1b-GFP, p3E-polyA, and pDestTol2AB2 using LR clonase. Final DNA constructs were microinjected with transposase RNA (30 ng/μl each) into one cell stage zebrafish embryos. In most cases, multiple independent stable alleles per construct were analyzed in the F1 generation (*Supplementary file 2*).

## In situ hybridization and immunohistochemistry

In situ hybridization and immunohistochemistry were performed as described (*Xu et al., 2018*). In addition to *foxc1a* and *foxc1b* probes (*Xu et al., 2018*), partial cDNAs were PCR-amplified with Phusion High-Fidelity DNA polymerase (New England Biolabs, Ipswich, MA), cloned into pCR_Blunt_II_-Topo (ThermoFisher Scientific, Waltham, MA), linearized, and then synthesized with Sp6 or T7 RNA polymerase (Roche Life Sciences, Indianapolis, IN) as specified (*Supplementary file 3*). For combined in situ hybridization and immunohistochemistry using *foxc1a* or *foxc1b* probes, we used rabbit anti-GFP antibody (TP401, Torrey Pines Biolabs, Secaucus, NJ) to detect *pou3f3b>GFP* followed by goat anti-rabbit secondary antibody (A11008, Invitrogen, Carlsbad, CA).

## Confocal imaging

In situ hybridization and transgenic embryos were imaged with a Zeiss LSM800 confocal microscope. Maximum-intensity projections are shown for stable transgenic lines, and single representative sections are shown for injected embryos. Image levels were modified consistently across samples in Adobe Photoshop CS6.

## μATACseq

Wild-type embryos double positive for *fli1a:EGFP* and *sox10:Dsred* (36 and 48 hpf), or *col2a1a:GFP* and *sox10:Dserd* (72 hpf), were sorted under a fluorescent dissecting microscope (Leica M165FC) before dissociation. For Foxc1 mutant analysis, we performed incrosses of *pou3f3b:Gal4+/−; UAS: nls-GFP+/−; sox10:Dsred+/−; foxc1a+/−; foxc1b+/−* fish and then selected for GFP+/Dsred+ embryos on a fluorescent dissecting microscope. Genotyping was then performed on tail lysates collected from individual embryos at 27 hpf. We then pooled *foxc1a−/−; foxc1b−/−* double mutants and separate sibling controls (*foxc1a+/−; foxc1b+/+*, and *foxc1a+/+; foxc1b+/+* embryos) for FACS. To facilitate embryo collection at the 36 hpf time point, embryos were moved at 27 hpf to an incubator set at 22°C to delay their development such that they reached 36 hpf the following morning. Cell dissociation and FACS were performed as previously described (*Askary et al., 2017*). Around 5,000 cells of each sample were centrifuged at 500 g for 20 m at 4°C, and the pellet was suspended with 20 μL of lysis buffer (10 mM Tris–HCl [pH 7.4], 5 mM MgCl$_2$, 10% DMF, 0.2% N-P40) by pipetting 6–10 times to release the nuclei without purification. The cell lysate was then mixed with 30 μL reaction buffer (10 mM Tris–HCl [pH 7.4], 5 mM MgCl$_2$, 10% DMF, and homemade Tn5 Transposase) by vortexing for 5 s. The reaction was incubated at 37°C for 20 min, followed by DNA purification using a Qiagen MiniElute kit. Purified DNA fragments were used to construct μATACseq libraries as previously described (*Buenrostro et al., 2013*) and sequenced using the NextSeq 500 platform (Illumina) with a minimum of 50 million paired-reads/sample. Two biological replicates of μATACseq experiments were performed for each condition.

## Data analysis and statistics

The Encode analysis pipeline (https://github.com/ENCODE-DCC/chip-seq-pipeline) for ATACseq (*Davis et al., 2018*) was used with small modifications. The raw reads were trimmed to 37 bp and aligned to the zebrafish GRCz10 genome assembly by STAR aligner (*Dobin et al., 2013*). PCR duplicates, and the reads that aligned to 'blacklist regions' (*Amemiya et al., 2019*), were removed, and then peaks were called by model-based analysis of ChIP-Seq (MACS2) (*Zhang et al., 2008*), with p=$10^{-7}$ cutoff and disabled dynamic lambda option (–nolambda) for individual replicates. Only peaks common for two biological replicates (Irreproducibility Discovery Rate < 0.1) were kept for further analysis. For visualization, bigwig files were generated from duplicates-removed bam files with *bedtools* and *bedgraphtobigwig*, and normalization was based on total read numbers. Heatmaps were generated with deeptools2 (*Ramírez et al., 2016*) based on the normalized bigwig signal files. Individual genomic loci were examined by IGV (Broad Institute). For quantitatively comparing the accessibility of distal regulatory elements between two conditions, peaks from two conditions were first merged, and only distal regulatory regions (1 kb upstream or 0.5 kb downstream from the transcription start sites) were kept for comparison. Raw read counts from each replicate were computed using 'bedtools multicov' function (bedtools multicov -bams input.bam -bed peak.bed > count.txt). Raw read count matrices from four replicates (two replicates per condition) were analyzed using DESeq2 package (*Love et al., 2014*). FDR = 0.1 was used as the cut off to filter the peaks that were

differentially accessible between two conditions. Volcano plots were generated using ggplot2 package in R. Fold change and adjusted p value were outputs from DESeq2 package in R, which is the same analysis behind the heatmaps in *Figure 3*. We used a $-\log10$ adjusted p value greater than one as the cutoff for determining significantly changed peaks. HOMER (*Heinz et al., 2010*) was used to identify de novo motifs and their associated p values. David 6.8 GO analysis was performed on the web interface (https://david.ncifcrf.gov/) based on nearest neighbor genes for all differentially accessible elements. Pearson correlation was calculated using cor() function in R.

## Materials and correspondence

Requests for material should be directed to J. Gage Crump (gcrump@usc.edu).

## Acknowledgements

We thank Jeffrey Boyd at the USC Stem Cell Flow Cytometry Core for FACS, David Ruble at the CHLA Sequencing Core, the high-performance computing core at USC, and Megan Matsutani, Jennifer DeKoeyer Crump, and Mathi Thiruppathy for fish care. We dedicate this work to the late Bartosz Balczerski whose passion as a postdoc inspired this project.

## Additional information

### Funding

| Funder | Grant reference number | Author |
| --- | --- | --- |
| National Institute of Dental and Craniofacial Research | R35 DE027550 | J Gage Crump |
| National Institute on Deafness and Other Communication Disorders | R01DC015829 | Neil Segil |

The funders had no role in study design, data collection and interpretation, or the decision to submit the work for publication.

### Author contributions

Pengfei Xu, Conceptualization, Formal analysis, Supervision, Investigation, Visualization, Methodology, Writing - original draft, Writing - review and editing; Haoze V Yu, Data curation, Formal analysis, Methodology, Writing - review and editing; Kuo-Chang Tseng, Data curation, Formal analysis, Methodology; Mackenzie Flath, Investigation; Peter Fabian, Resources; Neil Segil, Supervision; J Gage Crump, Conceptualization, Data curation, Formal analysis, Supervision, Funding acquisition, Writing - original draft, Project administration, Writing - review and editing

### Author ORCIDs

Kuo-Chang Tseng (iD) http://orcid.org/0000-0002-4870-7801
Neil Segil (iD) http://orcid.org/0000-0002-0441-2067
J Gage Crump (iD) https://orcid.org/0000-0002-3209-0026

### Ethics

Animal experimentation: This study was performed in strict accordance with the recommendations in the Guide for the Care and Use of Laboratory Animals of the National Institutes of Health. All of the animals were handled according to approved institutional animal care and use committee (IACUC) protocol (#20771) of the University of Southern California.

### Decision letter and Author response

Decision letter https://doi.org/10.7554/eLife.63595.sa1
Author response https://doi.org/10.7554/eLife.63595.sa2

# Additional files

## Supplementary files

• Supplementary file 1. Transcription factor binding motifs in cartilage-accessible elements. Detailed lists of predicted motifs for cartilage-accessible elements in 72 hpf controls (A), and based on timing of gain of accessibility (B), and dependence on Foxc1 function (C).

• Supplementary file 2. Summary of transgenic analysis of cartilage-accessible elements. Description of independent alleles for each enhancer transgenic line, genomic coordinates of elements tested, and activity in cartilage in 6 dpf zebrafish.

• Supplementary file 3. In situ probes. Details on primer sequences used to amplify probe regions for cloning, enyzmes used to linearize the probe plasmids, and the types of RNA polymerase used to synthesize RNA probes for in situ hybridization.

• Transparent reporting form

## Data availability

Chromatin accessibility data have been deposited in GEO under accession number GSE157575.

The following dataset was generated:

| Author(s) | Year | Dataset title | Dataset URL | Database and Identifier |
|---|---|---|---|---|
| Crump G | 2020 | Foxc1 establishes enhancer accessibility for craniofacial cartilage differentiation | https://www.ncbi.nlm. nih.gov/geo/query/acc. cgi?acc=GSE157575 | NCBI Gene Expression Omnibus, GSE157575 |

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
