## [Decision Letter]

**Acceptance summary:**

This paper examines the role of Foxc1 in mediating chromatin accessibility at specific enhancers to drive craniofacial cartilage development in zebrafish. It shows that cartilage-associated chromatin accessibility is dynamically established in neural crest cells in the zebrafish head, and that cartilage-associated regions that become accessible after neural crest migration are co-enriched for Sox9 and Foxc1 binding motifs. This work includes data sets that should prove valuable to the craniofacial community.

**Decision letter after peer review:**

Thank you for submitting your article "Foxc1 establishes enhancer accessibility for craniofacial cartilage differentiation" for consideration by *eLife*. Your article has been reviewed by three peer reviewers, and the evaluation has been overseen by a Reviewing Editor and Kathryn Cheah as the Senior Editor. The reviewers have opted to remain anonymous.

The reviewers have discussed the reviews with one another and the Reviewing Editor has drafted this decision to help you prepare a revised submission.

Summary:

This manuscript by Xu et al. examines the role of Foxc1 in mediating chromatin accessibility at specific enhancers to drive craniofacial cartilage development in zebrafish. Utilizing zebrafish transgenic lines to isolate specific populations of cranial neural crest cells (cNCCs), as well as more differentiated cNCC-derived chondrocytes at later stages of development, the authors perform several different sequencing experiments to identify cis-regulatory regions of accessible chromatin within cartilage specific genes.

Essential revisions:

The three reviewers agree that the data sets in your submission should prove very useful for the craniofacial community, and as such the study would merit publication as a short report following some revisions. In particular it was felt that some interpretations and conclusions in the paper were not well supported and should be attenuated.

1) The analysis/statistical methods in the paper need to be more detailed. There are no statistical methods clarified in the text or figure legends – please state what tests were applied, what the multiple sampling correction methods used were, etc.? Analysis methods are overall scarce and challenging to assess.

2) In order to draw the conclusion that Fox-dependent elements are enriched for both Fox and Sox motifs, combinatorial binding analysis needs to be carried out/included.

3) The conclusion that Foxc1 establishes cartilage-specific enhancer accessibility for Sox9's subsequent binding is unsupported by direct data and should be eliminated, and instead should be discussed as one possibility. Alternatively, ChiP data should be included that supports that conclusion.

4) The text should make clear that the Foxc1-dependent peaks could directly or indirectly require Foxc1 – there are currently no data in the manuscript to support that this is direct.

5) Some data should be included that validates that genes linked to the identified Foxc1-dependent exhancers show changes in expression in Foxc1 mutants – this could be done using RNA-Seq or qPCR.

6) The conclusion that Foxc1 mutant upper/dorsal face cartilage fails to develop is supported by images suggesting they were reduced relative to controls, but these data need to be quantified. Please also comment on the continued presence of Pou3f3b and Sox10 double-positive cells.

---

## [Author Response]

Essential revisions:The three reviewers agree that the data sets in your submission should prove very useful for the craniofacial community, and as such the study would merit publication as a short report following some revisions. In particular it was felt that some interpretations and conclusions in the paper were not well supported and should be attenuated.1) The analysis/statistical methods in the paper need to be more detailed. There are no statistical methods clarified in the text or figure legends – please state what tests were applied, what the multiple sampling correction methods used were, etc.? Analysis methods are overall scarce and challenging to assess.

We have added more extensive details in the “Data analysis and statistics” section of the Materials and methods. We note that P values for Homer motif and GO term analyses are determined within the relevant software packages. We now describe in the Legend to Figure 3—figure supplement 1 that data are shown as mean+/sem with p value calculated using a two-tailed students’ T test. We also now describe in the Methods and Legend to Figure 3—figure supplement 4 that a -log10 adjusted p value of 1 (equivalent to adjusted p value of 0.1) was used as the cut-off for significant peaks in the heatmaps and volcano plots.

2) In order to draw the conclusion that Fox-dependent elements are enriched for both Fox and Sox motifs, combinatorial binding analysis needs to be carried out/included.

We have now calculated the co-occurrence of Fox and Sox motifs and find that they are more than two-fold more likely in Foxc1-dependent than Foxc1-independent peaks.

“Whereas 33% (207/636) of Foxc1-dependent Group II elements had both Sox and Fox predicted motifs, only 16% (151/920) of Foxc1-independent Group II elements had both.”

“The co-enrichment of predicted Sox and Fox binding sites in 33% of chondrocyte enhancers suggests a model in which Foxc1 promotes Sox9 binding to these same enhancers by increasing chromatin accessibility…”

3) The conclusion that Foxc1 establishes cartilage-specific enhancer accessibility for Sox9's subsequent binding is unsupported by direct data and should be eliminated, and instead should be discussed as one possibility. Alternatively, ChiP data should be included that supports that conclusion.

We agree and have modified the text to indicate that, while our data support this model, direct evidence is lacking. We have removed statements in the Abstract and Results implying that Foxc1 opens enhancers for Sox9 binding and only mention this possibility in the Conclusion section.

“The co-enrichment of predicted Sox and Fox binding sites in 33% of chondrocyte enhancers suggests a model in which Foxc1 promotes Sox9 binding to these same enhancers by increasing chromatin accessibility, though this would require direct Sox9 binding studies to validate.”

4) The text should make clear that the Foxc1-dependent peaks could directly or indirectly require Foxc1 – there are currently no data in the manuscript to support that this is direct.

We agree and now better qualify that Foxc1 regulation could be direct or indirect.

“However, not all Foxc1-dependent enhancers have predicted Fox binding motifs, which might suggest indirect regulation of chromatin accessibility in at least some cases.”

“Further work will also be needed to understand the mechanism by which Foxc1 promotes chondrocyte enhancer accessibility, and the extent to which this reflects direct binding of Foxc1 to chondrogenic enhancers.”

5) Some data should be included that validates that genes linked to the identified Foxc1-dependent exhancers show changes in expression in Foxc1 mutants – this could be done using RNA-Seq or qPCR.

In new Figure 4C, we have performed in situ hybridization at 56 hpf and show selective reduction of expression of *sox10*, *lect1*, *col9a3*, and *epyc* in the dorsal arch cartilage-forming domains of Foxc1 mutants, similar to what we showed for *col2a1a* and *matn4* previously (Xu et al., 2018). Together, these findings demonstrate that at least 6 of the genes linked to Foxc1-dependent enhancers show corresponding decreases in expression in Foxc1 mutants.

6) The conclusion that Foxc1 mutant upper/dorsal face cartilage fails to develop is supported by images suggesting they were reduced relative to controls, but these data need to be quantified. Please also comment on the continued presence of Pou3f3b and Sox10 double-positive cells.

In new Figure 3—figure supplement 1, we now provide quantification showing a five-fold reduction in the size of dorsal cartilages in Foxc1 mutants. We also note that more extensive analysis of dorsal cartilage reduction in Foxc1 mutants was previously reported by us (Xu et al., 2018). Also in Figure 3—figure supplement 1, we have now performed *foxc1a* and *foxc1b* in situs in combination with anti-GFP staining to detect *pou3f3b>GFP*+ cells. These co-stainings show that *foxc1a/b* primarily overlap with *pou3f3b* in the dorsal-intermediate regions of the arches, with a *pou3f3b*-only domain in the dorsal-most region of the arches. We now comment in the text how this explains the continued presence of some *pou3f3b*+*/sox10*+ cartilage cells in the dorsal-most arch domains of Foxc1 mutants.

“Consistently, we observed reductions of cartilage in the *pou3f3b>GFP* domain of mutants at 6 days post-fertilization (dpf), except in the dorsal-most regions (close to the ear) that are *pou3f3b>GFP*-positive yet *foxc1a/foxc1b*-negative (Figure 3C, D; Figure 3—figure supplement 1B).”